# When ControlNet Meets Inexplicit Masks: A Case Study of ControlNet on its Contour-following Ability

## ABSTRACT

ControlNet excels at creating content that closely matches precise contours in user-provided masks. However, when these masks contain noise, as a frequent occurrence with non-expert users, the output would include unwanted artifacts. This paper first highlights the crucial role of controlling the impact of these inexplicit masks with diverse deterioration levels through in-depth analysis. Subsequently, to enhance controllability with inexplicit masks, an advanced *Shape-aware ControlNet* consisting of a deterioration estimator and a shape-prior modulation block is devised. The deterioration estimator assesses the deterioration factor of the provided masks. Then this factor is utilized in the modulation block to adaptively modulate the model's contour-following ability, which helps it dismiss the noise part in the inexplicit masks. Extensive experiments prove its effectiveness in encouraging ControlNet to interpret inaccurate spatial conditions robustly rather than blindly following the given contours, suitable for diverse kinds of conditions. We showcase application scenarios like modifying shape priors and composable shape-controllable generation. Codes are soon available.

## KEYWORDS

Text-to-Image Generation, ControlNet, Inexplicit Conditions, Shape-controllable Generation

## 1 INTRODUCTION

Text-to-Image (T2I) generation techniques [5, 23, 25, 26, 29, 31] have greatly changed the content creation area with high-fidelity synthesized images. By generating content following user-provided guidance like contours and shapes, ControlNet [37] stands out for its prominent capability of spatial control over T2I diffusion models and has become an indispensable tool for creation. The shape guidance always takes the format of segmentation masks, where the details of the contours and positions are perfectly reserved as shown in Fig. 1. However, such good *contour-following ability*[1] of ControlNet may cause artifacts when there exists noise in the mask, especially for non-expert users that have difficulty in providing accurate masks. Unfortunately, its performance on inexplicit masks with inaccurate contours remains under-explored, and studies on how to employ

[1]The contour-following ability refers to the ability to preserve the contours of conditional inputs in this paper.

**Unpublished working draft. Not for distribution.**

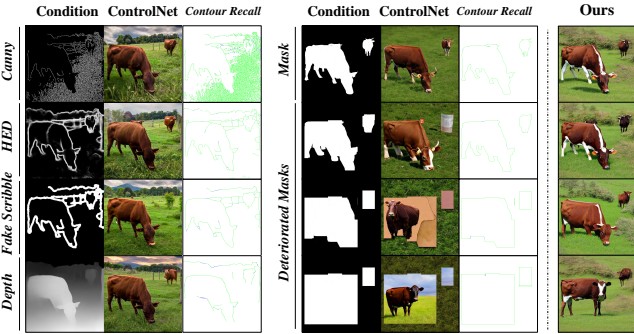

**Figure 1: ControlNet tends to preserve contours for spatial controllable generation over multi-modal control inputs, where green denotes recalled contours and blue denotes missing ones. However, inexplicit masks cause catastrophic degradation of image fidelity and realism. This paper largely enhances its robustness in interpreting inexplicit masks with inaccurate contours.**

Prompt: *A brown cow grazing on a patch of lush green grass*

inexplicit masks are ignored in current works. This issue hinders non-expert users from creating better images through ControlNet.

To better assist non-expert users in generating satisfactory images, this paper focuses on utilizing inexplicit masks during the generation process. We first analyze the most important property, *i.e.*, the contour-following ability, on the inexplicit mask-guided generation process through deteriorating masks and tuning hyperparameters. Specifically, we study this property quantitatively and reveal preliminary analyses on two key areas: 1) its performance on conditional masks of varying deterioration degrees, and 2) the influence of hyperparameters on this property. One of the main findings is that masks with inaccurate contours would cause artifacts due to strong contour instructions (§ 4.2). In other words, the contour-following ability implicitly assumes that *the conditional inputs align with the shape priors of specific objects*. Otherwise, it would cause artifacts or violate the spatial control in synthesized images as shown in Fig. 1. Given that precise conditional images are typically image-oriented (*e.g.*, human annotation or those extracted from reference images by offline detectors) or expert-provided (*e.g.*, artists), it constrains creation within the boundaries of existing images and experts. However, obtaining precise control inputs can be cumbersome and challenging for non-experts. This fact largely restricts creation through human scribbles, thus hindering broad applications of ControlNet.

While an intuitive solution is to adjust hyperparameters to achieve satisfactory results, finding the optimal setting can be challenging (§ 4.3). Besides, this strategy usually entails a trade-off between image fidelity and spatial control. Based on this observation, we improve ControlNet with the awareness of shape priors, namely Shape-aware ControlNet (§ 5), which alleviates the impact of strong contour instructions and shows advances in robust interpretation

of inexplicit masks (§ 6). Specifically, our improvements comprise a deterioration estimator and a shape-prior modulation block to automatically adjust the model's contour-following ability. The deterioration estimator assesses the deterioration ratio that depicts the similarity of the provided masks to explicit masks. Features from the Stable Diffusion (SD) [27] encoders are utilized during the estimation, as SD effectively encodes the shape priors of different objects. Then, we modulate this shape prior to the zero-convolution layers of ControlNet through the proposed shape-prior modulation block. This design enables extra control over the strength of contour instructions and adapts the model to the guidance of inaccurate contours. As Fig. 1 shows, our method follows object position and poses guided by inexplicit masks, keeping high fidelity and spatial control over T2I generation. It also adapts to other conditions beyond masks (§ 6.4). Moreover, though we only employ dilated masks for training, our method generalizes well to more realistic masks including user scribbles and programmatic TikZ sketches (§ 6.5).

This paper primarily focuses on ControlNet for two reasons. Firstly, ControlNet is the most representative and widely adopted method for spatially controllable T2I generation. It prevails over concurrent works for higher image quality and better preservation of outlines owing to the inherited powerful SD encoder and extensive training data. ControlNet is influential and has been widely incorporated in tasks including image animation [34, 36], video generation [8, 19], and 3D generation [4], *etc*. Secondly, ControlNet shares fundamental ideas and similar structures with existing adapter-based methods like T2I-Adapter [22]. Moreover, these methods exhibit similar properties (*i.e.*, the contour-following ability) and face similar issues (*i.e.*, performance degradation with inaccurate contours). More examples of ControlNet and T2I-Adapter are provided in Appendix Fig. S1. Therefore, our conclusions and methods can be potentially extended to these methods.

To summarize, our contributions are as follows:

- We study the contour-following ability of ControlNet quantitatively by examining its performance on deteriorated masks of varying degrees under different hyperparameter settings. We reveal inexplicit masks would severely degrade image fidelity for strong shape priors induced by inaccurate contours.
- We propose a novel deterioration estimator and a shape-prior modulation block to integrate shape priors into ControlNet, namely Shape-aware ControlNet, which realizes robust interpretation of inexplicit masks.
- Our method adapts ControlNet to more flexible conditions like scribbles, sketches, and more condition types beyond masks. We showcase its application scenarios like modifying object shapes and creative composable generation with deteriorated masks of varying degrees.

## 2 RELATED WORKS

*Training with Spatial Signals.* An intuitive solution to introduce spatial control is training models with spatially aligned conditions from scratch. Make-A-Scene [6] employs scene tokens derived from dense segmentation maps during generation, enabling complex scene generation and editing. SpaText [1] extends it to open-vocabulary scenarios and introduces spatio-textual representations for sparse scene control. Composer [16] decomposes images to representative factors like edges and then trains a model to recompose the input from these factors. However, training models from scratch requires large-scale training data and high computational costs.

*Adapters for Spatial Control.* Another kind of work focuses on the utilization of adapters to inject spatial control into the pretrained T2I models. For example, GLIGEN [18] involves gated self-attention layers to control the spatial layout during generation. T2I-adapter [22] and ControlNet encode the spatial guidance via lightweight adapters or duplicated UNet encoders, which are then fed into the decoder to inject spatial structures like shapes and contours. Uni-ControlNet [39], UniControl [24], and Cocktail [15] consolidate multi-modal conditions within a single framework. Since the base models are always frozen, these methods obtain better computational and data efficiency while retaining the generation ability of the base models. Despite these appealing characteristics, the impact of inexplicit masks is overlooked in these works, which may cause significant artifacts in generated images. Our proposed method can also be extended to these approaches to address this limitation.

## 3 PRELIMINARY

*ControlNet* [37] introduces a network as the adapter to control T2I generation models, *i.e.,* Stable Diffusion (SD), with extra spatially localized, task-specific conditional images including Canny edges, Hough lines, fake scribbles, key points, segmentation masks, shape normals, depths, and so on. Supposing one encoder block $F(\cdot; \theta)$ of SD parameterized by $\theta$, it accepts the input feature $x$ and output $y = F(x; \theta)$. ControlNet freezes the block and makes a trainable copy $F(\cdot; \theta')$ with parameter $\theta'$ to inject additional condition $c$ with zero convolution, formulated as,

$$y_c = F(x; \theta) + \lambda * Z(F(x + Z(c; w_1); \theta'); w_2), \quad (1)$$

where $Z(\cdot; w_1)$ and $Z(\cdot; w_2)$ are two $1 \times 1$ zero convolution layers with parameters $w_1, w_2$ initialized with zeros for stable training. $y_c$ is the output feature modulated by spatial control signals. $\lambda$, namely *conditioning scale*, is introduced during inference to adjust condition strength. Since features are additive, it allows compositions of multiple conditions, also known as Multi-ControlNet [37].

*Classifier-free Guidance (CFG)* [14] is a widely employed technique in generative diffusion models to increase image quality. Given a latent noise $z$, CFG is conducted through mixing samples generated by an unconditional model $\epsilon_\theta(\cdot)$ and a jointly trained conditional model $\epsilon_\theta(\cdot, c)$, which is computed as,

$$\tilde{\epsilon}_\theta(z, c) = \epsilon_\theta(z, c) + \omega * (\epsilon_\theta(z, c) - \epsilon_\theta(z)), \quad (2)$$

where $\tilde{\epsilon}_\theta(z, c)$ represents the final output under condition $c$. $\omega$ is a scaling factor for a trade-off between image quality and diversity, namely *CFG scale*. In this paper, the spatial conditions are injected into both $\epsilon_\theta(\cdot)$ and $\epsilon_\theta(\cdot, c)$ when conducting classifier-free guidance with ControlNet.

## 4 VISITING THE CONTOUR-FOLLOWING ABILITY

This section studies the contour-following ability from two aspects, including the performance on deteriorated masks of varying degrees and its interaction with hyperparameters, *i.e.*, CFG scale, conditioning scale, and condition injection strategy.

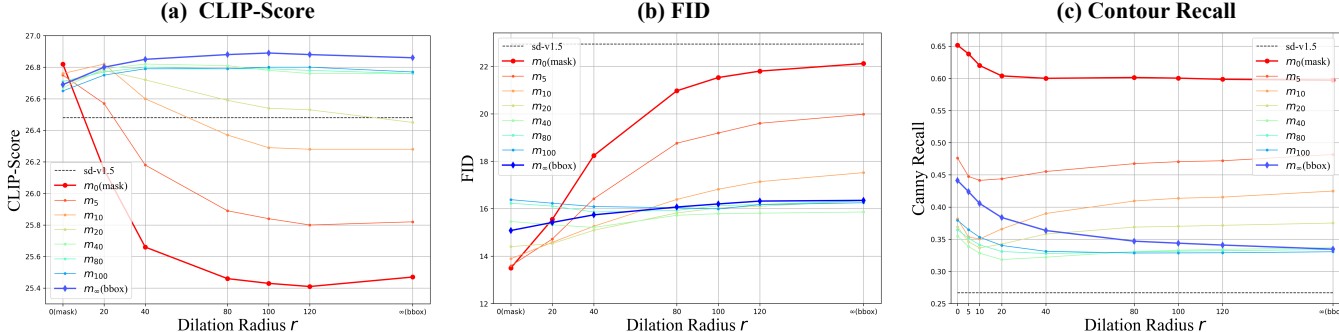

**Figure 2: The metric curves of ControlNet-$m_r$ on masks of varying deterioration degrees. The vanilla ControlNet, *i.e.*, ControlNet-$m_0$, suffers from dramatic degradation on CLIP-Score and FID on deteriorated masks, but keeps adhered to the contours of provided masks. ControlNet-$m_r$ ($r > 0$) exhibits more robust performance on deteriorated masks as the dilation radius $r$ becomes larger.**

## 4.1 Setup

We utilize the most common real-world control signal, *i.e.*, object mask for experiments. The object mask is simple but effective in covering shape information. For simplicity, we focus on plain binary masks in experiments, which are the most straightforward user-provided signals in real-world applications. This choice also helps to exclude the impacts of color and occlusion.

*Datasets.* Following Control-GPT [38], our experiments involve COCO images [20] and corresponding captions [3]. We utilize instance masks from LVIS [9], which offers sparse and precise human-annotated object masks of COCO images with over 1,200 object classes. We filter out images containing empty annotations, resulting in 114k image-caption-mask triplets for training and 4.7k for testing, namely the COCO-LVIS dataset.

*Implementation Details.* We adopt SD v1.5 [28] as the base model, and ControlNet is trained from scratch on COCO-LVIS for ten epochs with a learning rate of $1e - 5$ for all experiments. We take 50% prompt dropping for classifier-free guidance while keeping the SD parameters frozen. We use UniPC [40] sampler with 50 sampling steps. To evaluate, we take ground-truth masks for control and generate four images per caption. CFG scale and conditioning scale are set to 7.5 and 1.0 if there is no extra illustration.

*Metrics.* For evaluation, CLIP-Score (ViT-L/14) [11] and FID [12] are adopted as two basic metrics to measure the text-image alignment and image fidelity. In addition, as ControlNet tends to preserve all contour structures provided in the conditional images, we calculate the number of edge pixels retained in the generated images, namely *Contour-Recall (CR)*, to measure the contour-following effects quantitatively. Supposing a conditional image $c$ and generated images $X = \{x_i\}_{i=1}^N$, CR is defined as the recall of edge pixels in the control inputs, formulated as,

$$CR = \frac{1}{N} \sum_{i=1}^{N} \frac{|MaxPool(D(x_i), \sigma) \cap (D(c))|}{|D(c)|}, \qquad (3)$$

where $D(\cdot)$ is an edge detector that returns binary edge maps. We employ a max-pooling function $MaxPool(\cdot, \sigma)$ to tolerate an edge detection error of $\sigma$-pixels. In our implementation, we utilize the

Canny edge detector as $D(\cdot)$ and set a tolerance as $\sigma = 2$. A higher CR score indicates stronger contour-following ability.

## 4.2 Impact of Inexplicit Masks

To clarify concerns about ControlNet's performance on inexplicit masks, we first imitate inaccurate control signals by dilating object masks $m$ with progressively increasing radius $r (r \geq 0)$. The dilated mask is denoted as $m_r$, where $m_0$ represents masks with precise contours. Considering that bounding-box masks only convey object position and size without identifiable contours, we take an intersection over the dilated mask $m_r$ and the bounding-box mask denoted as $b$, to construct inexplicit masks of various degrees, formulated as,

$$m_r = Dilate(m, r) \cap b, \qquad (4)$$

where the bounding-box mask $b$ can also be expressed as the extreme case $m_\infty$ ($b \subset m_\infty$). In the rest of the paper, we use $m_\infty$ to denote the bounding-box mask. As $r$ grows large, the masks lose detailed shape information gradually.

We use the notation *ControlNet-$m_r$* to denote ControlNet trained with deteriorated masks $m_r$, where the vanilla ControlNet trained with precise masks is ControlNet-$m_0$. We test these models on masks of varying deterioration degrees to explore the performance of ControlNet with inexplicit masks and its contour-following ability. Fig. 2 presents the experimental results. Here are two main observations.

❶ **The vanilla ControlNet, *i.e.*, ControlNet-$m_0$, strictly adheres to the provided outlines, where inexplicit masks would cause severe performance degradation.** As shown in Fig. 1, the generated images are faithful to the contours of the control inputs under various control modalities, which is the key to excellent spatial controllable generation. One underlying reason is supposed to be that edges provide a shortcut to reconstruct the input images. Therefore, it exhibits strong contour instructions during the controllable text-to-image generation process of ControlNet.

However, we notice that such a contour-following ability is blind, which takes no account for inaccurate contours and maintains a high average CR of over 60% in Fig. 2. While it allows minor editing by manipulating masks to some extent, this property imposes severe problems on image realism when inaccurate contours (*e.g.* user scribbles) are provided. Similar observations are also noted in [2, 32, 33], but an in-depth analysis of this fact is neglected. For

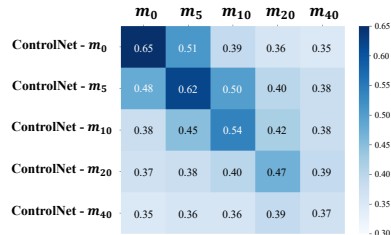

**Figure 3: Illustration of the inductive bias of *ControlNet-$m_r$* conditioned on $m_r$, where high CR on $m_0$ indicates models implicitly learn the dilation radius $r$.**

example, ControlNet tends to interpret rectangle masks as boards rather than animals specified in the prompt, causing incorrect spatial layouts and distorted objects. It is also reflected in a degradation of 1.35 CLIP-score and 8.62 FID in Fig. 2. Thus, precise masks with accurate contours are necessary for high-quality image generation. However, obtaining masks with accurate contours is challenging, especially for non-expert users.

❷ **Training ControlNet with deteriorated masks also converges and exhibits high robustness on masks of varying deterioration degrees with a sacrifice of contour instructions.** We further train a series of ControlNet, *i.e. ControlNet-$m_r$*, on progressively dilated masks $m_r$. Surprisingly, ControlNet converges with all conditions, even on the extreme bounding-box masks. As depicted in Fig. 2, training with deteriorated masks consistently improves ControlNet's robustness in interpreting inexplicit masks with better CLIP-score and FID at the cost of weak contour-following ability.

However, ControlNet trained with dilated masks implicitly assumes the presence of dilation in the provided mask, even for precise masks. Fig. 3 reports the CR referring to precise mask $m_0$ when ControlNet-$m_r$ is conditioned on dilated masks $m_r$. ControlNet-$m_r$ keeps a CR on $m_0$ over 0.5 until $r > 20$, suggesting that models infer and follow precise contours based on $m_r$. Visualized examples are provided in Appendix Fig. S2. This inductive bias diminishes as the radius $r$ increases, and ControlNet-$m_r(r \geq 40)$ shows weak contour-following ability and prompts object positions via masks.

Moreover, training ControlNet with severely deteriorated masks, *e.g.*, bounding-box masks, yields poorer CLIP-score and FID with precise masks as shown in Fig. 2. Examples in Fig. 7 illustrate that ControlNet-$m_\infty$ probably misinterprets one whole object mask into multiple small objects. This reminds us that too weak contour instructions can also cause problematic images and deviate from user intentions. So, it is worthwhile to develop a method to make contour-following ability more controllable, thereby fulfilling robust interpretation with inexplicit masks of varying degrees.

## 4.3 Impact of Hyperparameters

As a common practice, adjusting hyperparameters like CFG scales is a useful tool to achieve satisfactory generation results in T2I applications. Therefore, we investigate how hyperparameters impact the contour-following ability, especially when the conditional inputs are inexplicit, and reveal several guidelines.

Specifically, we test the vanilla ControlNet (*i.e.,* ControlNet-$m_0$), under two extreme condition cases, *i.e.*, precise mask $m_0$ and

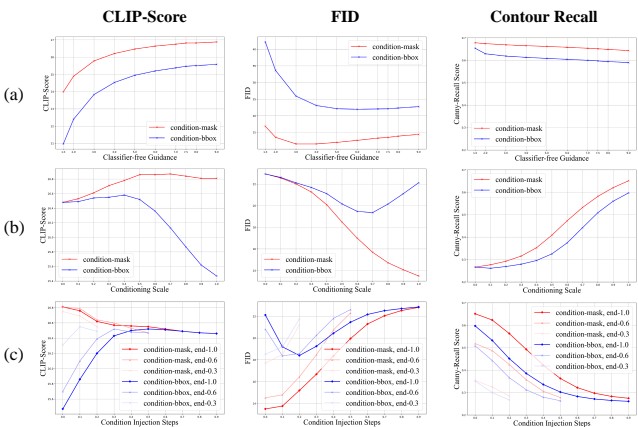

**Figure 4: Metric curves of (a) CFG scale, (b) conditioning scale, and (c) condition injection strategy for the vanilla ControlNet. Red denotes the performance on the precise mask $m_0$, and blue denotes the bounding-box mask $m_\infty$.**

bounding-box mask $m_\infty$. Three core hyperparameters are considered, *i.e.*, CFG scale $\lambda$, conditioning scale $\omega$, and condition injection strategy. Quantitative results are revealed in Fig. 4. For page limitations, visualized examples are presented in Appendix Fig. S3.

*Effects of Classifier-free Guidance $\omega$.* Under both conditions, ControlNet maintains high CR scores under all CFG-scale settings, showing little impact on contour-following ability. This is because conditioning signals are added to both $\epsilon_\theta(\cdot)$ and $\epsilon_\theta(\cdot, c)$ in Eq. 2. Moreover, performance on precise mask $m_0$ always surpasses that on $m_\infty$, indicating precise control inputs improve image quality. The trends of CLIP-Score and FID curves align with common sense, where a relatively higher $\omega$ brings better fidelity and text faithfulness.

*Effects of Conditioning Scale $\lambda$.* The conditioning scale decides the strength of control signals as Eq. 1. A higher $\lambda$ imposes stronger instructions, consistent with increasing CR scores under both conditions. However, the behavior diverges on CLIP-Score and FID curves when increasing $\lambda$. While both CLIP-Score and FID almost consistently increase on $m_0$, its performance on $m_\infty$ increases slightly at the very beginning but then declines dramatically, showing a large performance gap with $m_0$. This reveals a trade-off between image quality and spatial control with bounding-box masks, where $\lambda \in [0.5, 0.7]$ are recommended. Finding suitable $\lambda$ for masks of varying deterioration degrees usually requires tens of attempts.

*Effects of Condition Injection Strategy.* We divide the sampling steps into ten stages during the reverse sampling phase for experiments. As shown in Fig. 4(c), all stages contribute to contour instructions during the generation process in both cases. The first 10% ~ 40% steps hold the key, consistent with observations in T2I-Adapter [22]. In addition, a similar divergence of CLIP-Score and FID curves is observed between the two conditions, further declaring the performance gap between explicit and inexplicit masks.

In summary, while the CFG scale $\omega$ has little impact on the contour-following ability, a smaller conditioning scale $\lambda$ and discarding conditions at early reverse sampling stages help to relieve

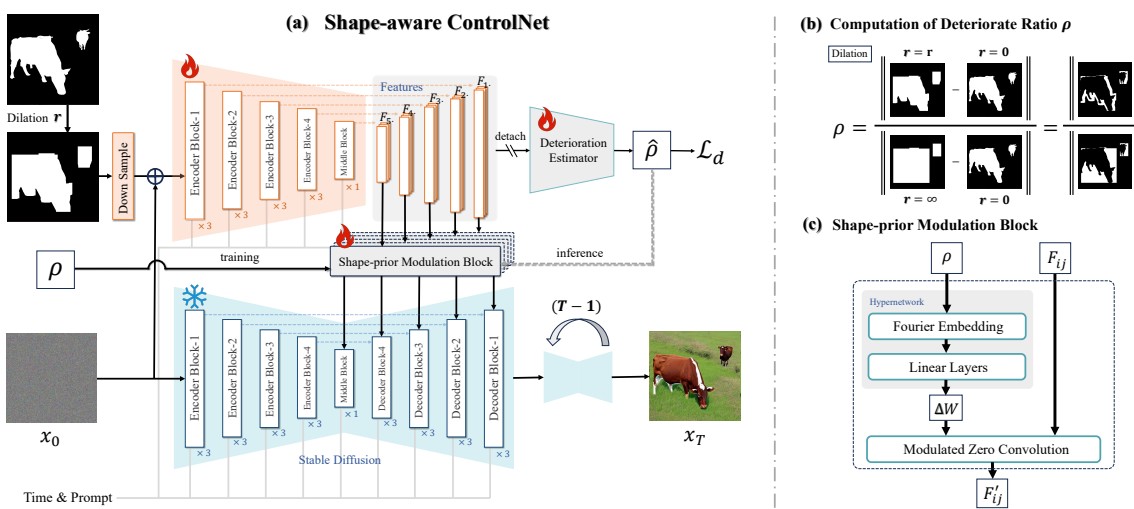

**Figure 5: The overall architecture of Shape-aware ControlNet. It contains 1) a deterioration estimator to assess the deterioration ratio of inexplicit masks, and 2) a shape-prior modulation block to modulate this ratio to ControlNet to adjust the contour-following ability for robust spatial control with inexplicit masks.**

contour instructions. These strategies help to relieve the degradation of image fidelity and text faithfulness for inexplicit masks and achieve a trade-off between image quality and spatial control. However, subtle changes to such hyperparameters may lead to dramatic changes in image appearance (see Appendix Fig. S3). Hence, it is still tricky to search for the ideal combination of hyperparameters for each image, and it also takes risks of violating spatial control.

## 4.4 Empirical Analysis

To clarify the difference between explicit and inexplicit mask controls, we compare ControlNet-$m_0$ with ControlNet-$m_\infty$ on precise masks $m_0$ and bounding-box masks $m_\infty$ in Fig. 6.

By removing textual prompts, we find that the strong contour-following ability of vanilla ControlNet derives intense shape priors from contours, thus it tends to infer specific objects from the mask directly. Since inexplicit masks introduce incorrect shape priors, they will cause conflicts with textual prompts or violations of spatial control for generation. For example, bounding-box masks are misinterpreted as the door in Fig. 6. This is why the vanilla ControlNet fails to interpret inaccurate contours with the correct content. VisorGPT [35] also validates that conditional generative models implicitly learn visual priors from the data, such as size and shape. If spatial conditions deviate from that prior, it would lead to artifacts and incorrect contents. In contrast, ControlNet-$m_\infty$ relies less on the shape priors from the conditional mask and prompts object locations for spatial control. This leads to improved text alignment and robustness on inexplicit masks. However, ControlNet-$m_\infty$ exhibits weak control with precise masks. So, it is necessary to develop a method to adapt ControlNet to deteriorated masks of varying degrees.

## 5 SHAPE-AWARE CONTROLNET

To enhance ControlNet's ability to interpret inexplicit masks robustly instead of blindly adhering to contours, we introduce a novel deterioration estimator and a shape-prior modulation block to ControlNet,

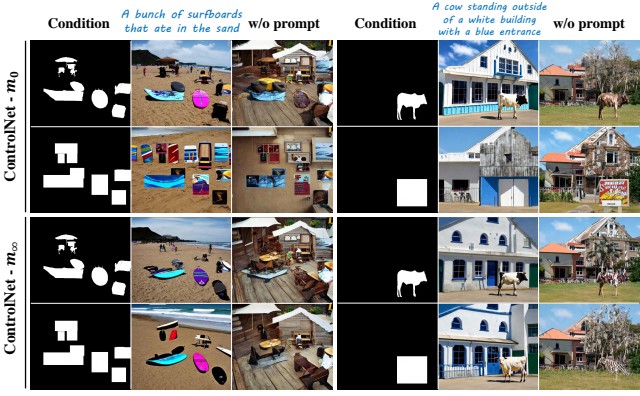

**Figure 6: Illustrations on the difference between explicit and inexplicit masks for control. The contour-following ability induces strong priors from contours, which causes conflicts with textual prompts and misinterprets conditional inputs.**

namely Shape-aware ControlNet, as depicted in Fig. 5. Our main insight is to introduce the shape prior explicitly to the model for better controllability.

## 5.1 Deterioration Estimator

We first introduce the deterioration ratio $\rho$, which depicts the gap between the inexplicit masks and explicit shape priors of specific objects, formulated as,

$$\rho = \frac{|S(\boldsymbol{m}_r) - S(\boldsymbol{m}_0)|}{|S(\boldsymbol{m}_\infty) - S(\boldsymbol{m}_0)|}, \tag{5}$$

where $S(\cdot)$ returns the area of masks, and $\rho \in [0, 1]$. Eq. 5 measures the shape priors of masks, where $\rho = 0$ indicates the exact object shape and $\rho = 1$ means no identifiable shape.

While computing the deterioration ratio $\rho$ based on precise object masks is straightforward, estimating $\rho$ from inexplicit masks is non-trivial. This motivates us to train an extra deterioration estimator. Since ControlNet-$m_r$ can imply the dilation radius $r$ (refer to § 4.2), we suppose ControlNet implicitly learns shape priors from deteriorated masks. So, we construct a deterioration estimator with stacked convolutional and linear layers with batch normalization, to predict the ratio from encoder features $\{F_{ij}\}$, where $F_{ij}$ denotes the $j$-th feature from the $i$-th encoder block. Such a design proves to provide accurate estimation as discussed in § 6.3. We train the estimator separately with an $L2$ loss and detach the gradients from the encoder to avoid negative effects on the convergence of ControlNet.

## 5.2 Shape-prior Modulation Block

Taking the shape prior $\rho$ as an extra condition, we design a shape-prior modulation block inspired by StyleGAN [17]. Specifically, we encode $\rho$ into Fourier embedding. Then, we employ a hypernetwork [10] to modulate shape priors to zero convolution layers in ControlNet as illustrated in Fig. 5(c). Supposing a zero convolution $Z(\cdot; w)$ parameterized by $w$ and ControlNet encoder feature $F_{ij}$, the modulated feature $F'_{ij}$ is computed as,

$$F'_{ij} = Z(F_{ij}; (1 + \Delta W) \cdot w), \tag{6}$$

where $\Delta W = H(\rho)$, and $H(\cdot)$ is a hypernetwork constructed by multiple linear layers with normalization.

## 6 EXPERIMENTS

### 6.1 Implementation Details

The baselines including ControlNet-$m_0$ and ControlNet-$m_\infty$ are implemented as § 4.1. We inherit the same setting to train our shape-prior modulation block on our COCO-LVIS dataset for 10 epochs from scratch with a learning rate of $1e-5$. The dilation radius $r$ is uniformly sampled from 0 to 80. The deterioration estimator is trained for another 10 epochs individually with gradient-detached features from ControlNet.

*Metrics.* We employ CLIP-Score and FID for evaluation. As the proposed *CR* metric would fail when the strict alignment of contours is violated, we propose two additional metrics inspired by SOA [13], namely Layout Consistency (LC) and Semantic Retrieval (SR). Definitions are as follows.

*Definition 6.1 (Layout Consistency, LC). LC measures the spatial alignment between conditional masks and generated images. Supposing detected bounding boxes $\{a_i\}_{i=1}^M$ and those of control masks $\{b_j\}_{j=1}^N$, LC is computed as follows:*

$$LC = \frac{|(\bigcup_i^M a_i) \cap (\bigcup_j^N b_j)|}{|(\bigcup_i^M a_i) \cup (\bigcup_j^N b_j)|}. \tag{7}$$

*Definition 6.2 (Semantic Retrieval, SR). SR is a retrieval measurement to verify whether the semantic objects assigned in the prompt are generated. Supposing M detected object categories $S = \{s_i\}_{i=1}^M$ over confidence threshold t and N assigned labels $L = \{l_j\}_{j=1}^N$, SR is formulated as,*

$$SR = \frac{|S \cap L|}{|L|}. \tag{8}$$

| Metric | Method | r | | | | | | AVG |
| | | 0 | 20 | 40 | 80 | 100 | $\infty$ | |
|---|---|---|---|---|---|---|---|---|
| CLIP-Score | ControlNet-$m_0$ | 26.82 | 26.15 | 25.66 | 25.46 | 25.43 | 25.47 | 25.83 |
| | ControlNet-$m_\infty$ | 26.69 | 26.80 | 26.85 | 26.88 | **26.89** | **26.86** | 26.83 |
| | **Ours** | **26.88** | **26.87** | **26.89** | **26.89** | 26.87 | 26.83 | **26.87** |
| FID | ControlNet-$m_0$ | 13.50 | 15.55 | 18.24 | 20.97 | 21.53 | 22.12 | 18.65 |
| | ControlNet-$m_\infty$ | 15.08 | 15.42 | 15.74 | 16.07 | 16.20 | 16.35 | 15.81 |
| | **Ours** | **13.20** | **13.62** | **14.07** | **14.72** | **14.78** | **15.12** | **14.25** |
| LC | ControlNet-$m_0$ | **0.522** | 0.440 | 0.374 | 0.327 | 0.320 | 0.303 | 0.381 |
| | ControlNet-$m_\infty$ | 0.401 | 0.430 | 0.438 | 0.444 | 0.445 | 0.446 | 0.434 |
| | **Ours** | 0.513 | **0.503** | **0.495** | **0.482** | **0.477** | **0.462** | **0.489** |
| SR | ControlNet-$m_0$ | **0.605** | 0.531 | 0.489 | 0.469 | 0.466 | 0.465 | 0.504 |
| | ControlNet-$m_\infty$ | 0.534 | 0.551 | 0.555 | 0.555 | 0.555 | 0.553 | 0.551 |
| | **Ours** | 0.601 | **0.586** | **0.577** | **0.569** | **0.567** | **0.561** | **0.577** |

**Table 1: Performance comparison of our method and baselines, *i.e.*, the vanilla ControlNet (ControlNet-$m_0$) and ControlNet-$m_\infty$ that trained with bounding-box masks, under different dilation radius $r$. Our method exhibits advances in interpreting deteriorated masks and achieves the best average performance.**

Specifically, we utilize an open-vocabulary object detector, *i.e.*, OWL-ViT [21] following VISOR [7]. We set the confidence threshold $t = 0.1$ and use instance labels as prompts.

### 6.2 Comparisons with the Vanilla ControlNet

As Tab. 1 depicts, our method achieves competitive or better performance on all metrics, demonstrating robustness under all dilation radius. Visualized results in Fig. 7 further illustrate the effectiveness of our method. ControlNet-$m_0$ tends to blindly follow the outlines of control inputs, thus suffering from unnatural images or violation of user intentions, especially when it is conditioned on largely deteriorated masks. On the other hand, ControlNet-$m_\infty$ struggles with precise masks with fine details like misinterpreting a single instance mask into multiple objects. In contrast, our method excels in understanding object orientations and shapes according to inexplicit masks, thanks to additional guidance of shape priors. More visualized examples are available in Appendix Fig. S4.

### 6.3 Ablation Studies

*Comparison with random dilation augmentation.* As augmentation with random dilated masks serves as an intuitive and straightforward solution to improve the robustness of deteriorated masks, we first compare this strategy with our Shape-aware ControlNet in Tab. 2. While this data augmentation strategy is also effective, our method achieves consistently higher performance on all metrics. Moreover, we highlight that our method provides users with the additional convenience of modifying the shape of generated objects through additional control over shape priors as presented in § 6.5.

*Accuracy of deterioration estimator.* We validate the accuracy of our proposed deterioration estimator in Fig. 8. The overall average $L1$ error under all dilation radius is 5.47%. Owning to the robustness of the modulation block, such an error shows minimal impact on the image fidelity as validated in Tab. 3. For page limitations, the complete error curves are presented in Appendix Fig. S5.

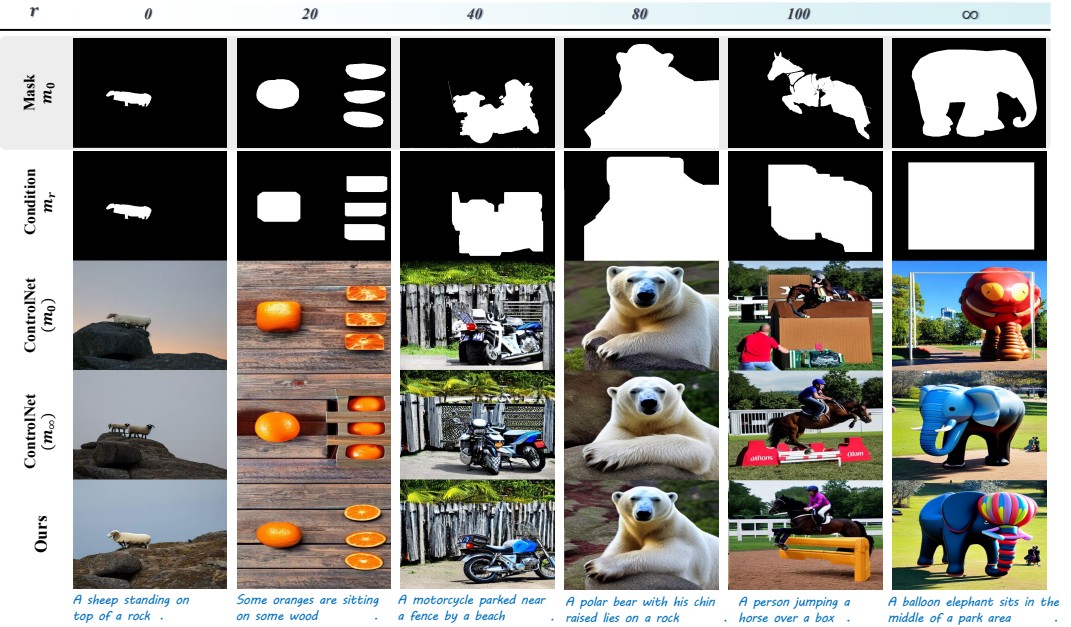

**Figure 7: Visualized comparison between the vanilla ControlNet and our Shape-aware ControlNet with masks at varying deterioration degrees. Our method not only follows explicit masks but also interprets inexplicit masks robustly.**

*Robustness on shape priors.* We further examine the robustness of our method on different shape priors. We take dilation radius $r = 20$ and manually adjust shape prior $\rho$ to $(\rho + \Delta\rho) \in [0, 1]$, where $\Delta\rho \in [0, 1]$. As shown in Fig. 9, changing $\rho$ reveals little impact on

| Method | CLIP↑ | FID↓ | LC↑(%) | SR↑(%) |
|---|---|---|---|---|
| ControlNet-$m_0$ | 25.83 | 18.65 | 0.504 | 0.381 |
| + Random Aug | 26.76 | 15.28 | 0.565 | 0.475 |
| Ours | **26.87** | **14.25** | **0.577** | **0.489** |

**Table 2: Comparison with random dilation for augmentation.**

| Ratio | CLIP↑ | FID↓ | LC↑(%) | SR↑(%) |
|---|---|---|---|---|
| $\rho$ | **26.88** | **14.21** | 0.576 | **0.489** |
| $\hat{\rho}$ | 26.87 | 14.25 | **0.577** | **0.489** |

**Table 3: Performance comparison between $\rho$ and predicted $\hat{\rho}$.**

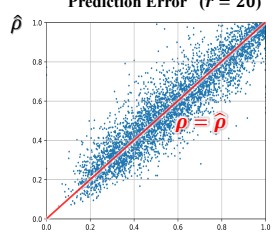

**Figure 8: Prediction error of deterioration estimator ($r = 20$).**

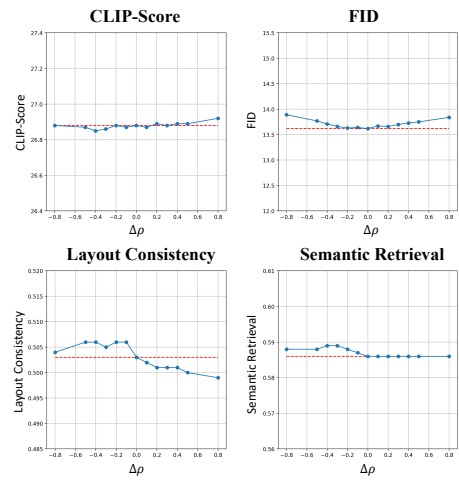

**Figure 9: The variation of metrics under $(\rho + \Delta\rho)$ when $r = 20$.**

the text faithfulness and image quality. But lower $\rho$ encourages better adherence to conditional masks, resulting in slightly higher LC with the assigned control masks, and vice versa. Visualized examples of tuning $\rho$ can be found in Fig. 12.

## 6.4 Extensive Results with More Conditions

It should also be noted that although the proposed method is designed for inexplicit masks, it can effortlessly adapt to other degraded conditions, demonstrating the versatility of our method in dealing with control signal deterioration. Here is an example with degraded edges and we leave other condition types for future works.

Specifically, we take the deteriorated edges from masks in Eq. 4 for condition. The experimental settings are the same as § 6.1 and the only change is the condition used for training. As depicted in Fig. 10, the vanilla ControlNet keeps adhering to the provided deteriorated edges blindly and struggles to generate high-quality images with inexplicit conditions, which are consistent with the observations on conditional masks. In contrast, our proposed method adapts well to degraded edges of varying degrees, proving its effectiveness in dealing with other types of degraded conditions. For page limitations, quantitative results are provided in Appendix Tab. S2.

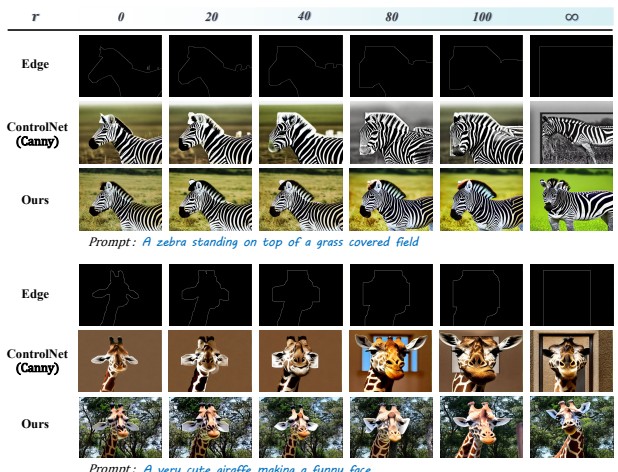

Figure 10: Comparison of ControlNet and our method with deteriorated edges. Our method can effortlessly adapt to diverse kinds of deteriorated conditions.

## 6.5 Applications

Here we showcase several applications of our method. For page limitations, additional examples are included in Appendix §A.

*Generation with TikZ sketches and scribbles.* While our model is only trained with dilated masks, we find it generalizes well to other types of realistic inexplicit masks and exhibits robust performance. Fig. 11 shows the effectiveness of our method in handling programmatic sketches and human scribbles. Our method generates reasonable images under abstract or inexplicit mask conditions while keeping high fidelity and spatial control. Here we follow Control-GPT [38] to prompt GPT-4 to produce TikZ codes for object sketches. Human scribbles are converted from Sketchy [30] dataset.

*Shape-prior modification.* While the deterioration estimator provides the deterioration ratio $\hat{\rho}$ for reference, we can manually set $\rho$ to control the shape prior. It determines how much the generated objects conform to the provided conditional masks as shown in Fig. 12. We notice it is possible to extend $\hat{\rho}$ to a large range of other values, even though $\rho \in [0, 1]$ for training. A lower $|\rho|$ empirically encourages stronger adherence to the contours of the provided masks.

*Composable shape-controllable generation.* We realize composable shape control via a multi-ControlNet structure [37], where we generate images by assigning different priors to each part of control

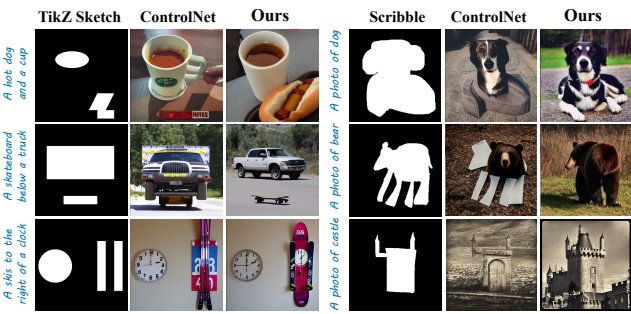

Figure 11: Performance comparison between our method and ControlNet on TikZ sketches and scribbles.

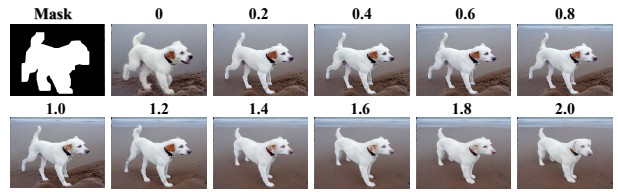

Figure 12: Shape-prior control via deterioration ratio $\rho$.

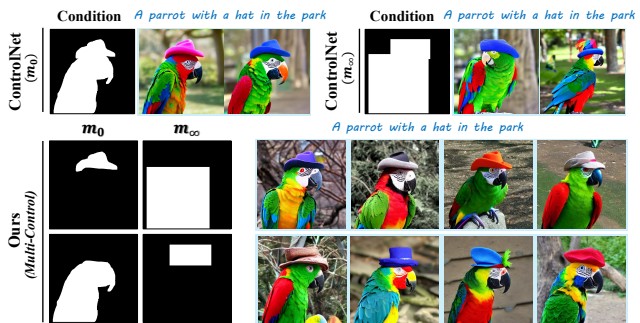

Figure 13: Examples of composable generation with explicit and inexplicit masks, showing flexible control with diverse results.

masks. This enables strict shape control over specifically assigned masks while allowing T2I diffusion models to unleash creativity in imagining objects of diverse shapes within inexplicit masks. A vivid example is presented in Fig. 13, where the parrots share fixed contours but hats are diverse in shapes, and vice versa.

## 7 CONCLUSION

In this paper, we reveal several insights into the core traits of ControlNet, *i.e.*, the contour-following ability. We quantitatively validate this property by distorting conditional masks and tuning hyperparameters, where we uncover severe performance degradation caused by inexplicit masks. In light of this, we propose a novel deterioration estimator and a shape-prior modulation block to endow ControlNet with shape-awareness to robustly interpret inexplicit masks with an extra shape-prior control. This exhibits the potential of employing ControlNet in more creative scenarios like user scribbles and diverse degraded condition types.

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
