# OpenReview forum: "When ControlNet Meets Inexplicit Masks: A Case Study of ControlNet on its Contour-following Ability"
_acmmm.org/ACMMM/2024/Conference — MM2024 Poster_

### Official Review · Reviewer_1XEk · 2024-05-09

**Rating:** 4
**Confidence:** 2

**Summary:**

The paper addresses a challenge faced by ControlNet. The authors highlight the need to manage the influence of such inexplicit masks through an in-depth analysis and propose an advanced version of ControlNet to enhance its robustness.
Shape-aware ControlNet introduces two key components: a deterioration estimator and a shape-prior modulation block. The deterioration estimator assesses the quality of the input masks, and the modulation block uses this assessment to adaptively adjust the model's contour-following ability. This adjustment allows the model to better handle noisy input, reducing the introduction of artifacts in the generated images.

**Strengths:**

- The proposed ControlNet is a novel enhancement method that has proven to be effective.
- The paper is well written and easy to follow.
- The paper includes many experiments and comparisons to assess the performance of the proposed method.

**Limitations:**

- How does the model determine when to follow a given contour and when not to? For example, in Figure 1, the user may wish for an object in the shape of a bucket to be repainted.
- Integrating additional components such as a deterioration estimator and a shape-prior modulation block could increase the computational complexity of the model. The computational costs and parameters associated with these modules are unclear.

**Suitability:**

3

---

### Official Review · Reviewer_94LY · 2024-05-23

**Rating:** 3
**Confidence:** 3

**Summary:**

This paper attempt to address the issue of ControlNet's inability to generate high-quality images when using rough masks as prompt conditions. This paper designed  a deterioration estimator to assess the deterioration factor. They also proposed a shape-prior modulation block to integrate shape priors into ControlNet. The experiments shows better results when the input masks are rough.

**Strengths:**

1. This paper addressed and, to some extent, resolved the current limitation of ControlNet's inability to generate high-quality images based on rough masks, which has practical implications.
2. The proposed framework made very lightweight modifications to ControlNet.

**Limitations:**

1. If the masks are relatively rough, whether it will be better to utilize the bounding box as the control signal, as providing bounding box is much easier. The advantage of rough mask over bounding box should be emphasized.
2. According to Figure 8, the prediction of deterioration is not precise, whether it means the prediction will not significantly influence the synthesized images. Whether the designed modules are useful for this task is not convincing.
3. In Table 2, the original ControlNet with random augmentation shows almost comparable performance with the proposed methods, whether that means the framework design is not that useful.

**Suitability:**

2

---

### Official Review · Reviewer_N3VU · 2024-05-27

**Rating:** 6
**Confidence:** 4

**Summary:**

This paper first highlights the crucial role of controlling the impact of these inexplicit masks with diverse deterioration levels through in-depth analysis. Subsequently, to enhance controllability with inexplicit masks, an advanced Shapeaware ControlNet consisting of a deterioration estimator and a shapeprior modulation block is devised. The deterioration estimator assesses the deterioration factor of the provided masks. Then this factor is utilized in the modulation block to adaptively modulate the model’s contour-following ability, which helps it dismiss the noise part in the inexplicit masks.

**Strengths:**

1. Sufficient experimental validation: The article validates the effectiveness of the proposed method through a large number of experiments, including the performance under masks with different degradation levels and the comparison with existing methods. These experimental results show the significant improvement of Shape-Aware ControlNet in image quality and spatial control.
2. In-depth theoretical analysis: The article not only proposes a new method, but also provides an in-depth quantitative analysis of ControlNet's contour-following ability, revealing its limitations when dealing with imprecise masks. This theoretical analysis provides a solid foundation for the proposed new method.
3. Promising applications: This paper demonstrates the potential of Shape-Aware ControlNet in several application scenarios, such as shape a priori modification and composable shape control generation. These application scenarios demonstrate the broad prospects of the method in practical applications and increase the likelihood of its acceptance.

**Limitations:**

1. Although the article conducts a large number of experiments, it mainly focuses on the COCO-LVIS dataset. The lack of validation on other datasets may affect the generalizability and persuasiveness of the method.

2. Certain technical details may not be described in sufficient detail, such as the specific implementation of the degeneracy estimator and the parameter settings of the shape a priori modulation module. This may affect the reader's understanding and reproduction of the method.

**Suitability:**

3

---

### Meta-Review · Area_Chair_zWd1 · 2024-07-05

**Recommendation:** Accept (Poster)
**Confidence:** 5

**Metareview:**

All reviewers are positive on this paper, and the authors have provided a strong rebuttal to clear concerns/questions. AC believes this paper deserves clear acceptance, and encourages the authors to further improve the paper when preparing camera ready.